# Biological Properties, Health Benefits and Enzymatic Modifications of Dietary Methoxylated Derivatives of Cinnamic Acid

**DOI:** 10.3390/foods10061417

**Published:** 2021-06-18

**Authors:** Magdalena Rychlicka, Agata Rot, Anna Gliszczyńska

**Affiliations:** Department of Chemistry, Wrocław University of Environmental and Life Sciences, Norwida 25, 50-375 Wrocław, Poland; rychlicka.magda@wp.pl (M.R.); 109265@student.upwr.edu.pl (A.R.)

**Keywords:** methoxylated derivatives of cinnamic acid, biological activity, bioavailability, lipids, phospholipids

## Abstract

Methoxylated derivatives of cinnamic acid play an important role in the formation of the pro-health potential of food products. Numerous reports present them as molecules with strong antimicrobial, antidiabetic, anticancer as well as hepato-, cardio-, and neuroprotective activities. In the last three decades, many research groups have tried to extend the practical application of these molecules as therapeutic and antioxidant agents extensively studying the methods of their lipophilization as the solution of problems of their low oral bioavailability and rapid metabolism. This article summarizes the latest data of natural sources of occurrence, biological potential and bioavailability of methoxy derivatives of cinnamic acids. Metabolism and pharmacokinetics of this group of dietary compounds are also extensively discussed as well as reviewing the methods of their chemical and enzymatic lipophilization in the aspect of their use in food and pharmaceutical industries.

## 1. Introduction

Polyphenols are a diverse group of plant secondary metabolites, which primary function is to protect plants against harmful ultraviolet UV radiation and pathogen attack [1]. In the plant kingdom, these biomolecules also play the role of factors that determine their color and taste as well as increase their attractiveness for pollinating insects [1]. Due to the ubiquitous presence of polyphenols in the plant world, they are an integral part of foods. Although these compounds do not exhibit nutritional functions in the body, numerous literature reports present them as valuable ingredients of the diet with health-promoting activities. Therefore, they are classified as bioactive phytochemicals that support the physiological functions of the body [2]. Polyphenols are also generally considered to be non-toxic to the human body. Because their lethal doses are very high, their oral overdose is extremely difficult to achieve [3].

Many literature data show that the estimated daily intake of polyphenols contained in fruits, vegetables, and beverages may reach approximately the level of 1700 mg [4]. This total dietary polyphenolic fraction consists of compounds belonging to flavonoids, stilbenes, lignans and phenolic acids [5]. The concentration of individual classes varies according to the source; however, it is worth noting that almost half of the daily dietary intake dose of polyphenols are phenolic acids (800 mg), mainly cinnamic acid derivatives (700 mg) and to a lesser degree, benzoic acid derivatives (100 mg) [4]. Therefore, the high content of cinnamates in natural products and in our diet indicates their significant contribution to the health-promoting potential of plants.

Phenylpropenoic acids commonly found in foods in the terms of chemical structure are hydroxy and methoxy derivatives of cinnamic acid. The most widespread like caffeic acid (CA), ferulic acid (FA), synapic acid (SA), *p*-coumaric acid (*p*-CA), *p*-methoxycinnamic acid (*p*-MCA) and 3,4-dimethoxycinnamic acid (3,4DMCA) are shown in Figure 1. These compounds are present in many food products, but berries, cereal, beverages and spices are considered as their richest sources [6,7,8]. The first one, caffeic acid (CA), is mainly present in coffee [9] and regular drinking of this beverage can supply from 250 to 500 mg of CA per day [10]. Coffee is also a valuable source of methoxy derivatives of cinnamic acid such as *p*-MCA and 3,4DMCA (690 mg/kg dry weight of coffee beans) [9]. Synapic acid (SA) and *p*-coumaric acid (*p*-CA) are compounds in which significant amounts have been determined in fruits [11]. Black mulberry fruits (1448 mg/kg dry weight) and strawberries (1107 mg/kg dry weight) are particularly rich sources of *p*-CA, while synapic acid has been found mainly in apples and pears, and its content can range from 15 to 600 mg/kg dry weight depends on their species [11]. Ferulic acid is commonly found in cereal raw materials such as corn, barley, wheat, rice and whole-grain bread where its content reaches up to 330 mg/kg of dry weight [12].

In most natural sources phenylpropenoic acids are rarely found in free form. Russell et al. reported that the concentration of their bounded form is even several hundred times higher than the concentration of their free forms [6]. This is the result of the role that these compounds play in plants, defend them against pathogen attack. A significant accumulation of phenylpropenoic acids is observed in the area of the plant cell walls. They occur there as the complexes with other compounds forming the scaffold of biological structures which are resistant to chemical and physical factors. This significantly limits the bioavailability of these acids by humans because in this form they are not directly absorbed from the gastrointestinal tract. Thus, their absorption is possible only after release by hydrolysis processes catalyzed mainly by intestinal microflora enzymes [6].

Many of the phenylpropenoic acids including (especially) the methoxy derivatives of cinnamic acid—ferulic acid, *p*-methoxycinnamic acid and 3,4-dimethoxycinnamic acid—are also active compounds of medicinal plants such as: angelica sinensis (*Angelica sinensis*), common columbine (*Aquilegia vulgaris*), hogweed (*Cimicifuga heracleifolia*), leprechaun (*Scrophularia buergeriana*) or kencur (*Kaempferia galanga*). These plants have been used for centuries in Eastern medicine as natural remedies for preventing and treating many diseases related to the function of the digestive, respiratory, nervous and immune systems [3,11,12,13,14,15,16].

## 2. Biological Activity of Methoxylated Derivatives of Cinnamic Acid

The biological activity of compounds highly depends on their chemical structure, which affects such parameters as solubility, ability to penetrate biological membranes or binding to the active center of enzymes. In the case of natural derivatives of cinnamic acid, their bioactivity is mainly determined by the presence of hydroxy (-OH) and methoxy substituents (-OCH_3_) in the aromatic ring, which number and site of substitution define the wide range and potency of their action.

One of the first reports in the literature on the relationship between structure and biological potential of methoxy derivatives of phenylpropenoic acids dates back to the 1970s, when Japanese scientists published a report on the strong antioxidant activity of ferulic acid [17]. They proved that this phenolic acid exhibits chain-breaking activity and prevents ischaemia–reperfusion-associated intestinal injury [18]. A few years later, ferulic acid was accepted as an antioxidant food additive in Japan [19]. In the following years, it also became the subject of worldwide experiments, which confirmed its antimicrobial [20] and anticancer (against lung [21], breast [22], cervical [23], prostate [24], thyroid [25] and gastrointestinal [26] cancers) properties in in vitro tests. Furthermore, such beneficial activities as antidiabetic [27] hepato- [28], cardio- [29] and neuroprotective [30] activities were also confirmed during the in vivo tests in animal models. Of particular interest seems to be the anti-atherosclerotic activity of ferulic acid, which during studies in mice, showed a more potent effect than reference compound—clofibrate, a drug commonly used to reduce blood cholesterol level. It was reported that FA significantly increased the activity of hepatic and erythrocyte antioxidant enzymes (superoxide dismutase, catalase, glutathione peroxidase, glutathione reductase, and paraoxonase) that inhibit the oxidation of low-density lipoprotein (LDL), and thus showed potent anti-atherosclerotic activity [29]. Currently, a ferulic acid-based preparation is available in China and has found application in the treatment of cardiovascular diseases. However, due to the low bioavailability of its active ingredients, very high doses of this preparation are necessary to be administrated to achieve a therapeutic effect [31].

Comprehensive studies on the relationship between the presence of a methoxy group in the aromatic ring of phenylpropenoic acids and their biological activity have been conducted by the Adisakwattana team between 2004 and 2017. As a result of a series of in vitro and in vivo tests, the researchers observed that the presence of the -OCH_3_ group in the *para* position is a key structural element that determines the high antidiabetic activity of phenylpropenoic acids. It was proven that *p*-methoxycinnamic acid exhibits 100-fold higher activity than the reference compound 1-deoxynojirimycin with documented antidiabetic properties [32]. The authors of this study explained that such high activity of *p*-MCA acid is a result of multidirectional action, which includes stimulation of insulin secretion, improvement of pancreatic *β*-cell function, delay of carbohydrate digestion and glucose uptake, inhibition of protein glycation, insulin fibrillation and gluconeogenesis in the liver [33,34,35,36]. Similar observations regarding the antidiabetic properties of acids were also described by other study authors [37,38].

Other in vitro studies demonstrate also the potent hepatoprotective properties of methoxylated derivatives of cinnamic acid in tests performed on rat’s hepatocytes with toxicity induced by carbon tetrachloride (CCl_4_). These studies showed that from among tested phenylpropenoic acids, those with the -OCH_3_ group in the *para* position (such as *p*-methoxycinnamic acid and isoferulic acid) exhibit hepatoprotective activity comparable to that of sylibine, even when used in several dozen times-lower concentrations. It has been confirmed that compounds belonging to this group have a direct effect on the activity of liver enzymes and thus positively influence the oxidative stress balance, improve lipid and alcohol metabolism, inhibit inflammation, fibrosis and apoptosis of liver cells [28]. These results have been further confirmed during in vivo studies conducted on the mice and rat model [16,28,39,40].

As it turns out, methoxy derivatives of cinnamic acid and their ethyl esters also exhibit significant anti-amnestic properties [30,41,42,43,44]. The results of in vivo studies conducted on the mouse model have shown that *p*-MCA is able to reduce memory deficits by about 60% in scopolamine-induced amnesia rodents, compared to the control group. It is worth noting that its effect is superior to that of a substance called velnakrine used in clinical trials as a drug in Alzheimer’s disease [43]. Furthermore, 3,4-dimethoxycinnamic acid was also found to be active in neuroprotective properties. Under in vitro studies in human neuroblastoma cells line (SH-SY5Y), 3,4DMCA strongly bound with prion proteins, reducing the possibility of oligomer formation of these proteins by 30–40% compared to control and significantly increasing the viability of these cells [44].

The data presented in Table 1 indicate a number of anticancer properties of methoxy derivatives of cinnamic acid [45,46,47,48,49,50,51,52]. Of particular interest seems to be the anticancer activity of *p*-MCA acid, which against colon cancer cells (HCT-116) is similar to doxorubicin. At this point, it should also be highlighted that the activity of *p*-MCA was more selective against HCT-116 tumor cells than against the normal colon epithelial cell line (NCM460) [49].

Summary of biological activities of methoxy derivatives of cinnamic acids like: ferulic, *p*-methoxy- and 3,4-dimethoxycinnamic acids and their ester derivatives (ferulic acid methyl ester (MFA), *p*-methoxycinnamic acid ethyl ester (E*p*-MCA)) with their mechanisms of action are presented in Table 1.

Summing up, it can be concluded that extensive research has demonstrated that the chemical structure of phenylpropenoic acids highly defines the extent of their therapeutic potential. While the presence of the -OH group in the aromatic ring of phenylpropenoic acids is mainly responsible for the occurrence of antioxidant activity, the group -OCH_3_, especially in the *para* position relative to the unsaturated side chain, determines their strong antidiabetic as well as hepato- and neuroprotective potential. This proves the high therapeutic potential of this group of compounds both in the prevention and therapy of civilization diseases.

## 3. Bioavailability of Phenylpropenoic Acids

The therapeutic effect and possibility of the practical application of phytochemicals as therapeutic agents significantly depends on their bioavailability, understood as the concentration in which the compounds reach the general circulation in an unchanged form. This, in the first line, depends on their solubility and permeability through the gastrointestinal tract. One of the main parameters affecting the degree of absorption of cinnamic acid derivatives in the human body is their chemical structure. Kern et al. carried out a study to determine the bioavailability of ferulic acid (FA) from food [53]. For this purpose, they determined the concentration of this compound that appeared in the blood of volunteers after consumption of a meal consisting of cereal bran, containing 22.5 μM FA per kg of their body weight. The maximum concentration of the compound (expressed as the sum of its metabolites) that was observed in peripheral blood was only 0.2 μM/L 180 min after consumption of the meal. However, the total amount determined in urine, the main route of FA excretion from the body, was about 3% in relation to the amount ingested with food. Such a low content of FA in plasma is caused by its limited absorption from the gastrointestinal tract due to its chemical form of occurrence in natural sources. As it was mentioned earlier, phenolic acids rarely occur in natural sources in the free form. Most of these compounds occur in combinations with mono-, di-, and polysaccharides, sterols, polyamines, glycoproteins, and lignins that form the structure of plant cell walls. Such combinations are not metabolized by human digestive enzymes. Release of free acids and consequently also absorption occurs only as a result of hydrolytic enzymes of intestinal microflora [53]. A diagram showing the absorption of dietary methoxy derivatives of phenylpropenoic acids in the human body is presented in Figure 2.

A relatively higher degree of absorption of phenolic acids can be obtained by their administration in the form of salts or esters of short-chain alcohols. However, even then, the chances of achieving satisfactory concentrations in peripheral blood are still low due to the rapid metabolism of these compounds in the system [54,55]. During the studies in rats, it was observed that orally administered free FA is predominantly absorbed from the stomach (74%), from where it reaches the liver through the portal vein. There, as a result of enzymes phase I and II of xenobiotic metabolism, it undergoes sulfonation and conjugation with glucuronic acid to form derivatives with lower biological activity. As a result of such intensive transformations, there is no unchanged form of ferulic acid in the bloodstream and it is almost immediately excreted via urine. Rapid metabolism of ferulic acid was also confirmed in human studies. Orally administrated ferulic acid sodium salt in a dose of 50 mg caused a maximum plasma concentration of 2.5 μmol/L of its unchanged form, observed after 24 min after administration. The calculated half-life of this molecule in the human body was then only 42 min [55].

The metabolic processes of phenylpropenoic acids also depend on their structure, especially on the number of hydroxy and methoxy substituents present in the aromatic ring. During in vitro studies in human colorectal adenocarcinoma cells (Caco-2), it was proven that 3,4-dimethoxycinnamic acid penetrates the intestinal wall several times easier than the corresponding hydroxy derivative of cinnamic acid [56]. Moreover, an experiment performed using post-mitochondrial supernatant from mammalian liver cells (S9 fraction) also showed that due to methylation of hydroxy groups, this acid is also characterized by higher metabolic stability [56]. The presence of -OCH_3_ groups inhibits the activity of first-pass liver enzymes responsible for sulfonation and glucuronidation reactions. Therefore, methoxylated derivatives of cinnamic acid reach the bloodstream in unchanged form. However, despite the fact that *O*-methylated forms of phenolic acids more easily penetrate into the blood, animal studies show that these compounds are still characterized by a short half-life which is less than 1 h [57].

## 4. Enzymatic Modifications of Methoxylated Derivatives of Cinnamic Acids with Lipid Molecules

In recent years, several strategies aimed at improving the pharmacokinetic properties of phenolic compounds in biological systems have emerged. One of them is based on achieving increase bioavailability of this group of compounds by using lipid carriers. Lipids as hydrophobic compounds, after passing through the intestinal wall, penetrate into the lymphatic system through which they join the bloodstream at the level of the thoracic vein. This route of transport of lipids in the circulatory system distinguishes them from hydrophilic compounds which, after passing through the intestinal walls, enter the blood vessels, and then are transported by the portal vein to the liver and only then to the general circulation. Therefore, assigning a lipid character to phenolic acids could allow increasing their concentration in the system by increasing the degree of absorption, bypassing the “first pass mechanism through the liver” and then prolonging the time of release and circulation in the blood [58]. Currently, there are already preparations available on the market, which use the strategy of lipid carrier of the active substance. An example is Siliphos^®^, a complex of phosphatidylcholine and silibin, which after oral administration shows 5-fold higher bioavailability compared to free silibin [59]. This type of solution for phenylpropenoic acid was proposed by Kusumawati and Yusuf in their study. In order to increase the activity of ethyl *p*-methoxycinnamate (E*p*-MCA) they prepared its phospholipid complex. Studies carried out on mice confirmed that oral administration of this preparation resulted in a 2-fold increase in the concentration of ester in the system [60].

Although the delivery of active substances via lipid carriers is effective, this method also possesses some limitations. Due to the fact that in most cases, this method is based on the encapsulation of hydrophilic compounds in the aqueous core or on the entrapment of hydrophobic molecules in the lipid bilayer, premature “leakage” of the active substance is often observed as the results of an action of physical and chemical factors [61]. One strategy that can solve this problem is the formation of a direct or indirect (through a linker compound) covalent bond between the biologically active molecule and the lipid compound. This type of solution gives the possibility to release the active substance only as a result of the action of endogenous enzymes present in the target tissues [60]. Clinical studies have shown that therapeutic compounds administered in the form of lipid–drug conjugates (LDCs) are characterized by higher bioavailability after oral administration and lower toxicity compared to free drug forms. In addition, the release of the drug from such combinations can be controlled, reducing the occurrence of potential side effects [61]. In recent years, the Food and Drug Administration (FDA) and the European Medicines Agency have approved the first drugs produced in the form of lipid conjugates, which have found applications especially in the areas of treatment of diabetes, schizophrenia and depression [62].

The covalent combinations of phenolic acids with neutral lipids like fatty alcohols and acylglycerols have already been widely described in the literature what was summarized in Table 2. Such derivatives of methoxylated phenylpropenoic acid have found applications mainly in the cosmetic industry. For instance, Laszlo et al. developed a process for the enzymatic transesterification of soybean oil with ferulic acid ethyl ester in a reaction catalyzed by Novozym 435. The product of this reaction was a mixture of ferulated mono- and diacylglycerols registered under the trade name SoyScreen as an ingredient used in the manufacture of creams with a filter actively absorbing UVA/UVB radiation [63]. Lipophilic compounds with UV-absorbing ability were also obtained by enzymatic esterification reaction *p*-MCA with 2-ethylhexanol developed by two independent teams [64,65]. Biological studies also showed that the obtained ester has higher antioxidant activity than ascorbic acid and the free form of *p*-MCA, as well as good antimicrobial activity against a number of pathogenic microorganisms [65,66]. In 2006, also Weber’s research group reported the results of enzymatic synthesis of long-chain alkyl esters of *p*-methoxycinnamic acid, which can be used as lipophilic additives with antioxidant activity in food, feed and cosmetic industries [67].

The most recent strategy for obtaining phenolipids is based on the formation of their conjugates with phospholipids, mainly with phosphatidylcholine (PC). The feature that distinguishes PC from other lipids is its amphiphilic character which determines its unique physicochemical properties such as high compatibility with biological membranes. Moreover, PC is a valuable source of choline and polyunsaturated fatty acids (PUFA) in our diet. It has been demonstrated during human studies that more than 90% of phosphatidylcholine is absorbed from the small intestine, and its maximum blood concentration of 20–30% of the ingested dose is observed about 6 h after ingestion [82]. These studies demonstrate that phosphatidylcholine has unprecedentedly high absorption from the gastrointestinal tract and intense circulation in the bloodstream.

In recent years, in our research team, we reported studies on the chemical synthesis of phosphatidylcholines containing in their structure *O*-methylated phenolic acid as acyl fragments. In vitro studies confirmed for these conjugates higher antiproliferative activity towards selected human cancer cell lines: MV4-11 (leukemia), A549 (lung cancer), MCF-7 (breast cancer), LoVo (colorectal cancer), LoVo/DX (drug-resistant colorectal cancer) and HepG2 (liver cancer) than for free forms of corresponding acids [83,84]. Additionally, their selectivity of action was also confirmed in tests carried out towards normal cells of mouse fibroblasts line Balb/3T3. Subsequent studies revealed also their higher ability to induction of glucose-stimulated insulin secretion and intracellular calcium mobilization in the MIN6 β pancreatic cell line being at the same time not toxic for these cells at active doses [85]. Balakrishna et al. also described increased antioxidant potential obtained as a result of chemo-enzymatic synthesis of phospholipids structured with ferulic, synaptic, vanillic and syringic acid [86], whereas Anakanbil et al. successfully applied chemo-enzymatic synthesis pathway to obtain 1-acyl-2-caffeoyl and 1-caffeoyl-2-acyl PC [87]. A two-step enzymatic reaction of the synthesis of lysophosphatidylcholine containing ferulic acid was proposed by Yang [88]. In this report, the researchers first enzymatically hydrolyzed natural soy phosphatidylholine and then transterified obtained lysophosphatidylcholine with ethyl ferulate in a reaction catalyzed by Novozym 435. Ferulated lysophospholipids were also further successfully produced during one-step lipase-catalyzed transesterification of egg yolk PC with higher reaction yield [89]. Moreover, in the following years, methoxylated phenolic acids such as *p*-methoxy [90], 3,4-dimethoxy [91] and anisic acid [92] were successfully introduced into phospholipid structure via Novozym 435-catalyzed acidolysis or transesterification reaction (Figure 3).

Analyzing the process of enzymatic production of phospholipids containing methoxylated derivatives of cinnamic acid, it is observed that in the first step, 2-acyl-lysophosphatidylcholine (2-acyl-LPC) is formed by hydrolysis of native PC. The regioselectivity of this reaction, towards the *sn*-1 position, was confirmed by the results of GC analyses. The acid profile in the obtained phospholipid fractions (containing natural (PC, LPC) and structured phospholipids (modified PC, modified LPC)) revealed a decrease in the concentration of saturated fatty acids, which present mainly in the *sn*-1 of natural PC [91]. The hydroxy group of phosphatidylcholines released in this way undergoes the next transesterification with an acyl residue donor molecule, yielding the first reaction product which is modified phosphatidylcholine. However, it should be noted that the formation of this product occurs with very low efficiency (usually less than 5%) and in the case of modification with EF, it was not observed at all. The probable reason for this phenomenon is the successive conversion of 2-acyl-LPC to 1-acyl-LPC by spontaneous migration of fatty acids from *sn*-2 to *sn*-1 position. Thus, the formed 1-acyl-LPC is again hydrolyzed by Novozym 435 to glycerophosphocholine (GPC), which is an intermediate product and simultaneously a substrate for the synthesis of modified lysophosphatidylcholines. The presence of GPC in the product mixture was observed by TLC and HPLC analyses.

## 5. Conclusions and Future Perspective

Methoxylated derivatives of cinnamic acid are important bioactive dietary compounds, for which, in many in vitro and in vivo studies, high pro-health impact has been demonstrated. The evidence reviewed above supports the notion that methoxylated derivatives of cinnamic acid exhibit higher biological activity than their corresponding hydroxyderivatives. In this regard, the methoxy group at the *para* position of the benzene ring is crucial, which was especially highlighted in the case of antidiabetic and anticancer activities. Moreover, met oxylated derivatives of cinnamic acid are more stable in biological systems and the presence of the methoxy group prevents the formation of glucuronic acid and sulfate conjugates. However, still, it does not change the fact that they undergo rapid metabolism and occur in low concentration in the free form in natural sources by which their bioavailability in the human body is very low and the achievement of therapeutical effects is very difficult. Therefore, over the years, many attempts have been made to develop the methods of their lipohilization, which is detailed analyzed in the paper.

## Figures and Tables

**Figure 1 foods-10-01417-f001:**
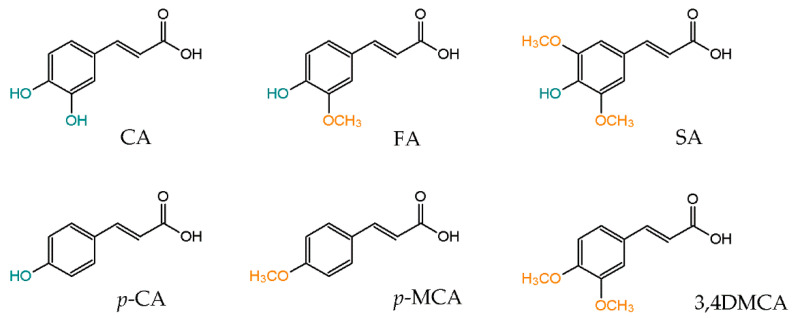
Chemical structure of dietary phenylpropenoic acids.

**Figure 2 foods-10-01417-f002:**
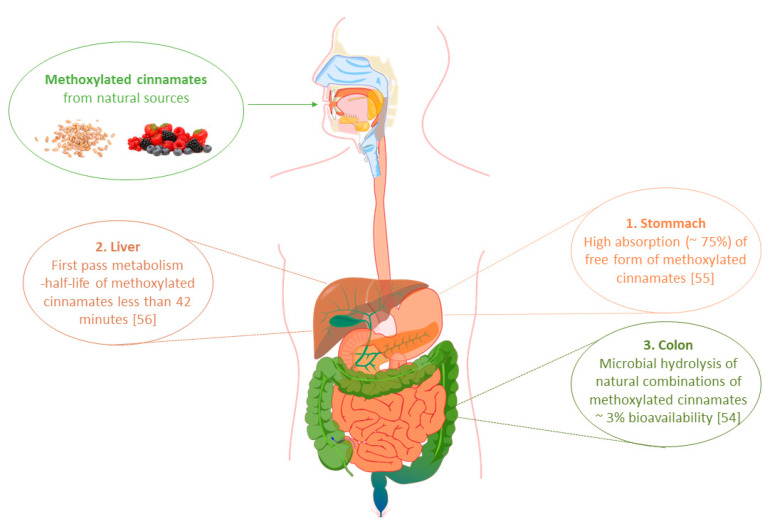
Absorption of dietary methoxylated derivatives of phenylpropenoic acids in the human gastrointestinal tract.

**Figure 3 foods-10-01417-f003:**
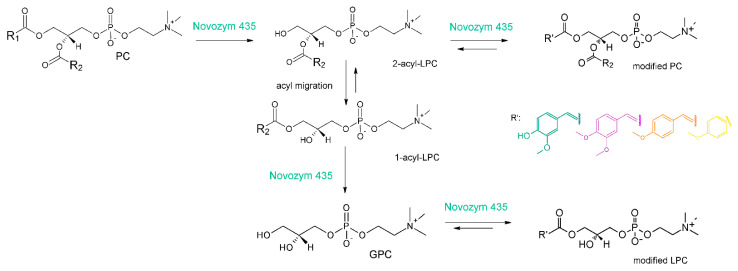
Possible pathway of the lipase-catalyzed synthesis of phenylated lysophospholips via acidolysis/transesterification reaction.

**Table 1 foods-10-01417-t001:** Biological potential and mechanism of action of methoxy phenylpropenoic acids.

Activity	Molecule	Research Model	Active Dose	Mechanism of Action	Refs.
Hepatoprotective	FA	rats/EtOH	20 mg/kg b.w.	ALP↓, GGT↓, ALT↓, AST↓	[28]
	MFA	mice/EtOH	5–20 mg/kg b.w.	ALT↓, AST↓, SOD↑, CAT↑, GSH↑ GSH-Px↑, T-AOC↑	[40]
	*p*-MCA	rats/CCl4	1–5 μM	GSH↑, GSH-Px↑, GST↑, GR↑	[16]
	*p*-MCA	rats/CCl4	50 mg/kg	ALP↓, ALT↓, GGT↓	[39]
Antydiabectic	*p*-MCA	INS-1 cell line	100 μM	insulin↑	[34]
	*p*-MCA	INS-1 cell line	100 μM	ions Ca2+↑, insulin↑	[35]
	*p*-MCA	Wistar rats	5 mg/kg	insulin↑	[34]
	*p*-MCA	rats/STPZ	40 mg/kg	insulin↑, glukogenesis↓	[36]
	*p*-MCA	rats/STPZ	40–100 mg/kg	insulin↑	[37]
	*p*-MCA	rats/STPZ	10–40 mg/kg	Insulin↑	[38]
Neuroprotective	FA	rats-ROT	50 mg/kg m.c.	neuron necrosis↓, IL-1β↓, IL-6↓, TNF-α↓, CAT↑, SOD↑	[30]
	*p*-MCA	rat cortical cells/glutamate	1 μM	-	[41]
	E*p*-MCA	rat cortical cells/glutamate	0.01–1 μM	ions Ca2+↑, glutamatergic antagonism	[42]
	E*p*-MCA	mice-ICR	0.01–2 mg/kg	-	[43]
	3,4-DMCA	SH-SY5Y cell line	400 nM (Kd)	cell viability↑	[44]
Chemopreventive and anticancer	FA	MDA-MB-231 mices	100 mg/kg	apoptosis↑ metastatic potential↓	[23]
	FA	mice/UV radiation	50 mg/kg	DNA protection	[45]
	FA	PC-3, LNCaP cell line	300 μM 500 μM	cell proliferation↓	[25]
	FA	rats/DMAB	40 mg/kg	-	[46]
	FA	rats/4NQO	500 ppm	-	[47]
	*p*-MCA	HepG2 cell line	27.1 g/mL (IC50)	apoptosis↑	[14]
	*p*-MCA	rats	40 mg/kg	-	[48]
	*p*-MCA	HCT-116 cell line	10 μM (IC50)	apoptosis↓	[49]
	E*p*-MCA	mice/DMAB	23.4 mg/kg	apoptosis↓	[50]
	E*p*-MCA	MCF-7 cell line	360 g/mL (IC50)	-	[51]
	E*p*-MCA	rats/DMH	40 mg/kg	-	[52]

↓/↑—decrease/increase in activity, - — not known mechanism of action, ALP—alkaline phosphatase, GGT—γ-glutamyl transferase, ALT—alanine transaminase AST—aspartate transaminase, SOD—superoxide dismutase, CAT—catalase, GSH—glutathione, GSH-Px—glutathione peroxidase, T-AOC—total antioxidant capacity, GST—glutathione S-transferase, ROT—rotenone, IL-1β, IL-6—proinflammatory interleukins, TNF-α—tumor necrosis factor, PC-3 and LNCaP—human prostate cancer cell lines, MDA-MB-231, MCF-7—human breast cancer cell lines, HepG2—human liver cancer cell line, HCT-116—human colon cancer cell line DMAB—7,12-dimethylbenz(a)anthracene, STPZ—streptozotocin, 4NQO—nitrohinone oxide, DMH—1,2-dimethylhydrazine.

**Table 2 foods-10-01417-t002:** Lipase-catalyzed modification of methoxylated derivatives of cinnamic acid with neutral fatty acids and triacylglycerols.

Type of Lipid	Molecule	Type of Reaction	Reaction Conditions	Conv.	Refs.
**FATTY** **ALKOHOLS**	FA	esterification	*Rhizomucor miehei*, 50 °C, 12 days, octanol	30%	[68]
	FA	esterification	Novozym 435, 92.2 °C, 3 days, octanol	93%	[69]
	FA	esterification	Novozym 435, 78 °C, 1 day,2-ethylhexanol	100%	[70]
	FA	esterification	Novozym 435, 60 °C, 4 days, oleyl alcohol in ionic liquide/isooctane octanol	97%	[71]
	FA	esterification	*Rhizomucor miehei*, 50 °C, 3 days, decyl alkohol in *n*-hexan	88%	[72]
	EF	transesterification	Novozym 435, 60 °C, 6 days, octanol	83%	[73]
	*p*-MCA	esterification	Novozym 435, 80 °C, 1 day, 2-ethyl hexanol	90%	[64]
	*p*-MCA	esterification	*Rhizopus oryzae*, 45 °C, 4 days, cyclooctan, 2-ethylhexanol	91%	[65]
	*p*-MCA 3,4MCA FA	esterification	Novozym 435, 80 °C, 2.5 days cis-9-octadecen-1-ol	94% 25% 94%	[67]
	M*p*-MCA MF EF	transesterification	Novozym 435, 80 °C, 3 days, cis-9-octadecen-1-ol	92% 91% 89%	[74]
	3,4DMCA	esterification	Novozym 435, 60 °C, 15 days, octanol	3%	[75]
	EF	transesterification	Novozym 435, 45 °C, 6 days, soy oil	64%	[63]
**TRIACYLGLICEROLS**	EF	transesterification	Novozym 435, 60 °C, 22 days, cocconut butter shea butter,	63% 70%	[76]
	EF	transesterification	Novozym 435, 50 °C, 5 days, tributyryne in toluene	75%	[77]
	EF	transesterification	Novozym 435, 90 °C, 7 days, castor oil	100%	[78]
	FA	acidolysis	Novozym 435, 55 °C, 10 days, trioleine in hexane/butanone, (85/15)	62%	[79]
	FA 3,4DMCA	acidolysis	Novozym 435, 55 °C, 10 days, linseed oil in hexane/butanone, (85/15)	19% 10%	[80]
	FA	acidolysis	Novozym 435, 55 °C, 10 days, trioleine in hexane/butanone, (75/25)	15%	[81]

FA—ferulic acid, EF—ethyl ferulate, MF—methyl ferulate, *p*-MCA—*para* methoxy cinnamic acid, E*p*-MCA—ethyl *p*-met—oxycinnamate, M*p*-MCA—methyl *p*-methoxycinnamate, 3,4MCA—3,4-dimethoxycinnamic acid.

## Data Availability

Data are contained within the article.

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
