# Peer review of "Biological Properties, Health Benefits and Enzymatic Modifications of Dietary Methoxylated Derivatives of Cinnamic Acid"

_foods, 2021, doi:10.3390/foods10061417_

Round 1

Reviewer 1 Report

This manuscript was description the role of partial methoxylated and hydroxyl derivatives of cinnamic acid relative to health potential of food products. The author through biological activity, bioavailability, and enzymatic modifications with lipid molecules to discuss the deriver of cinnamic acid. The comment of this manuscript is reconsider after major revision. The following specific comments should be considered while revising the manuscript.

  1. 95-97. The sentence is not clear, please rewrite.
  2. Table 1 shows that methoxylated derivatives of cinnamic acid has four biological activities, including hepatoprotective, antydiabectic, neuroprotective, chemopreventive and anticancer. However, some of the cited references are not mentioned in the text. The cited references listed in Table 1 should be divided into these four functions to be explained and added in L.151.
  3. In the Table 1, the row of chemopreventive and anticancer, DMAB is the abbreviation of 7,12-dimethylbenz anthracene in ref. 47, but in the foot note of table 1 is DMAB - 3,4-dimethyl,4-aminobiphenyl. Please check.
  4. 208. add comma after “intensive transformations, there…….”
  5. The following references are related to lipase-catalyzed esterification of ferulic acid, and add these references to Table 2 to make Table 2 more complete.

Highly efficient esterification of ferulic acid under microwave irradiation. Molecules, 2009, 14(6), 2118-2126.

Developing a high-temperature solvent-free system for efficient biocatalysis of octyl ferulate. Catalysts, 2018, 8(8), 338.

Highly efficient synthesis of an emerging lipophilic antioxidant: 2-ethylhexyl ferulate. Molecules, 2016, 21(4), 478.

  1. 235 and L. 352, The section number should be 4 and 5.
  2. 264, add comma “Due to the fact that in most cases, this method”
  3. 318. add comma “syringic acid [85], whereas Anakanbil…”
  4. 337. add comma “of cinnamic acid, it is observed that…”
  5. 343-344. The word “modified phosphatidylcholine” is not clear. Does this refer to the product in the upper right corner of the Fig.3? Please mark it in Fig. 3.
  6. 350. The word “modified lysophosphatidylcholines” is not clear. Does this refer to the product in the bottom right corner of the Fig.3? Please mark it in Fig. 3.
  7. I did not see the point of this manuscript to fit title (Biological properties, health benefits and enzymatic modifications) in the conclusion, please rewrite the conclusion.

Author Response

Prof. Dr. Paweł Kafarski

Prof. Dr. Izabela Jasicka-Misiak

                                                                                               Guest Editors of Foods

Dear Editor in Chief of Foods,

We would like to thank you for the opportunity to revise our paper on “Biological properties, health benefits and enzymatic modifications of dietary methoxylated derivatives of cinnamic acid” (foods-1252162). The offered suggestions have been immensely helpful for us. We have included the Reviewers comments and responded to them individually, indicating exactly how we addressed each concern and describing the changes we have made. All changes were highlighted in yellow in revised version of the manuscript. As the Reviewer indicates the English corrections, we checked the manuscript paying more attention on grammar and spelling. The manuscript was also checked by the native speaker for language inconsistencies as well.

According to the Reviewer 1 comments:

  1. 95-97. The sentence is not clear, please rewrite.

Response: We agree with the Reviewer that the sentence could be not clear and we have already rewrite it.

  1. Table 1 shows that methoxylated derivatives of cinnamic acid has four biological activities, including hepatoprotective, antydiabectic, neuroprotective, chemopreventive and anticancer. However, some of the cited references are not mentioned in the text. The cited references listed in Table 1 should be divided into these four functions to be explained and added in L.151.

Response: Yes, Reviewer has right that not all cited references listed in Table 1 were mentioned in the text. We have added the appropriate citations in the text and divided it into subsections according to described biological activity.

  1. In the Table 1, the row of chemopreventive and anticancer, DMAB is the abbreviation of 7,12-dimethylbenz anthracene in ref. 47, but in the foot note of table 1 is DMAB - 3,4-dimethyl,4-aminobiphenyl. Please check.

Response: Thank you, it was error. Now it is corrected.

  1. add comma after “intensive transformations, there…….”

Response: Thank you, we did it.

  1. The following references are related to lipase-catalyzed esterification of ferulic acid, and add these references to Table 2 to make Table 2 more complete.

Highly efficient esterification of ferulic acid under microwave irradiation. Molecules, 2009, 14(6), 2118-2126.

Developing a high-temperature solvent-free system for efficient biocatalysis of octyl ferulate. Catalysts, 2018, 8(8), 338.

Highly efficient synthesis of an emerging lipophilic antioxidant: 2-ethylhexyl ferulate. Molecules, 2016, 21(4), 478.

Response: Listed by the Reviewer references regarding the enzymatic modifications have been already added to Table 2.

  1. 235 and L. 352, The section number should be 4 and 5.
  2. 264, add comma “Due to the fact that in most cases, this method”
  3. 318. add comma “syringic acid [85], whereas Anakanbil…”
  4. 337. add comma “of cinnamic acid, it is observed that…”

Response: Thank you for corrections, we confirm that we made them.

  1. 343-344. The word “modified phosphatidylcholine” is not clear. Does this refer to the product in the upper right corner of the Fig.3? Please mark it in Fig. 3.
  2. 350. The word “modified lysophosphatidylcholines” is not clear. Does this refer to the product in the bottom right corner of the Fig.3? Please mark it in Fig. 3.

Response: Trying to make this part more clear we added on the Figure 3 subscription which compound is modified PC and which one is modified LPC moreover we have changed also mentioned sentence.

  1. I did not see the point of this manuscript to fit title (Biological properties, health benefits and enzymatic modifications) in the conclusion, please rewrite the conclusion.

Response: According to the Reviewer suggestions we changed the conclusions section.

According to the Reviewer 2 comments:

  1. Figure 1: improve resolution if possible

Response: According to the Reviewer suggestions we attached the new version of Figure 1 with higher resolution.

  1. Consider adding a materials and methods section with some details on review strategy, coverage, search terms etc

Response: After careful analysis we decided not to add the other section trying not to prolong more the review.

  1. Line 101-104 and throughout: somehow it is difficult to understand if the effects are based on in vitro studies only or if there is human clinical evidence or at least in vivo animal experiments

Response: We fully agree with the Reviewer that this part was difficult for analysis. Therefore, we divided this sentence into two indicating which activities were tested only in the in vitro model and which one in the in vivo model.

  1. General comment: could you add a section on adverse effects, if unusually high levels of these compounds are ingested, or supra-physiological levels using the increased bioavailability products

Response: Phenylpropanoids are reported in the literature as non-toxic compounds. Expressed for them lethal doses are very high so oral overdose is extremely difficult to be achieved. We added this information to the section 1.

  1. Regulatory comment: would the compounds with increased bioavailability need an approval as “novel food product” before being able to be placed on the market?

Response: Reviewer has right conjugates of p-MCA and other phenylpropanoids require first approval before we can call them “novel food product” so we can only tell that they can be reconsidered as potential novel food products.

  1. Line 322: typo in …choline

Response: Thank you, we corrected this.

We would like to express once again our thanks to Reviewers for very valuable comments, which help us to improve our manuscript. We hope that our explanations and corrections are sufficient, and will be accepted.

With kind regards,

Anna Gliszczyńska

Reviewer 2 Report

The  authors review the properties and health effects of some cinnamic acid derivatives.

The review is extremely focused on the medicinal and therapeutic properties of the compounds, which somehow often reach far beyond what would be acceptable for a food coming into the realm of medicinal products. For the journal “Foods”, the food characteristics, such as use as antioxidant or preservative, should be stressed. Otherwise, more appropriate journals would be available at MDPI for medicinal products/pharmaceutics.

Some detailed comments:

  • The English should be checked by a native speaker
  • Figure 1: improve resolution if possible
  • Consider adding a materials and methods section with some details on review strategy, coverage, search terms etc
  • Line 101-104 and throughout: somehow it is difficult to understand if the effects are based on in vitro studies only or if there is human clinical evidence or at least in vivo animal experiments.
  • General comment: could you add a section on adverse effects, if unusually high levels of these compounds are ingested, or supra-physiological levels using the increased bioavailability products
  • Regulatory comment: would the compounds with increased bioavailability need an approval as “novel food product” before being able to be placed on the market?
  • Line 322: typo in …choline

Author Response

(The authors gave the same response as above.)

Round 2

Reviewer 1 Report

The article has been revised according to the suggestions, and can be accepted after the suggestion below are revised.

L. 33-34. Add comma. Because their lethal doses are very high, their oral overdose is extremely difficult to achieve [3].

L. 98-100. Is it appropriate to change the sentence like this “they proved that due to the presence of a hydroxy substituent at the para position of ferulic acid forms a phenoxy radical is stabilized by resonance interactions.”

L.372. Add comma. “compounds, for which in many in vitro and in vivo studies, high pro-health……”

L.539. Reference 63, the authors should be Huang, K. C., Li, Y., Kuo, C. H., Twu, Y. K., & Shieh, C. J.

Author Response

14 May, 2021

Prof. Dr. Paweł Kafarski

Prof. Dr. Izabela Jasicka-Misiak

Guest Editors of Foods

Dear Editor in Chief of Foods,

We would like to thank you for the opportunity to make additional changes in our paper on “Biological properties, health benefits and enzymatic modifications of dietary methoxylated derivatives of cinnamic acid” (foods-1252162). All changes were now highlighted in green in new version of revised manuscript.

According to the Reviewer 1 comments:

  1. 33-34. Add comma. Because their lethal doses are very high, their oral overdose is extremely difficult to achieve [3].

Response: Thank you, we added the comma.

  1. 98-100. Is it appropriate to change the sentence like this “they proved that due to the presence of a hydroxy substituent at the para position of ferulic acid forms a phenoxy radical is stabilized by resonance interactions.”

Response: Yes, we agree with the Reviewer that it is too general comment and we rewrite this sentence.

  1. 372. Add comma. “compounds, for which in many in vitro and in vivo studies, high pro-health……”

Response: Thank you, now it is corrected.

  1. 539. Reference 63, the authors should be Huang, K. C., Li, Y., Kuo, C. H., Twu, Y. K., & Shieh, C. J.

 Response: Thank you very much for indication the error. Now the reference 63 is correct.

We would like to express once again our thanks to Reviewer 1 for his very valuable comments, careful revision and help.  We hope that our corrections are now sufficient, and will be accepted.

With kind regards,

Anna Gliszczyńska
